# Detection of *Venturia inaequalis* Isolates with Multiple Resistance in Greece

**DOI:** 10.3390/microorganisms10122354

**Published:** 2022-11-28

**Authors:** Michael Chatzidimopoulos, Antonios Zambounis, Fenia Lioliopoulou, Evangelos Vellios

**Affiliations:** 1Department of Agriculture, Crop Production and Rural Environment, University of Thessaly, 38446 N. Ionia, 38221 Volos, Greece; 2Institute of Plant Breeding and Genetic Resources, Hellenic Agricultural Organization ‘ELGO DIMITRA’, 57001 Thessaloniki, Greece

**Keywords:** *Venturia inaequalis*, fungicide resistance, trifloxystrobin, cyprodinil, dodine, difenoconazole, boscalid, fludioxonil, dithianon, captan

## Abstract

The excessive use of fungicides against *Venturia inaequalis*, the causal agent of apple scab, has led to the emergence of resistant populations to multiple fungicides over the years. In Greece, there is no available information on fungicide resistance, despite the fact that control failures have been reported on certain areas. An amount of 418 single-spore isolates were collected from three major apple production areas and tested for their sensitivity to eight commonly used fungicides from unrelated chemical groups. The isolates were tested on malt extract agar media enriched with the discriminatory dose of each fungicide using the point inoculation method. To define the discriminatory dose for assessing the levels of resistance, EC_50_ values on both spore germination and mycelial growth assays were previously determined. Isolates exhibiting high resistance to trifloxystrobin (92% in total) and difenoconazole (3%); and moderate resistance to cyprodinil (75%), dodine (28%), difenoconazole (36%), boscalid (5%), and fludioxonil (7%) were found for the first time in Greece. A small percentage of the isolates were also found less sensitive to captan (8%) and dithianon (6%). Two isolates showed various levels of resistance to all eight fungicides. Despite the occurrence of strains with multiple resistances to many fungicides, we concluded that this practical resistance in the field arose mainly due to the poor control of apple scab with trifloxystrobin and difenoconazole.

## 1. Introduction

To date, the control of apple scab caused by *Venturia inaequalis* (Cooke) Winter, relies heavily upon fungicides because it is widely accepted that consumers typically prefer flawless fruit lacking any signs of parasite infection. To achieve this goal, the common farmer practice is to avoid primary infections caused by ascospores with fungicide sprays in 5–8 day intervals [1]. The spraying programs in Greece usually start at the green-tip stage of the crop in March–April and end at the end of May when the leaves have become less susceptible to infections [2]. Furthermore, simulation models may assist in spray decisions by enabling the better timing of spray applications according to the infection risk periods; however, such models are not adopted in all apple growing areas [1]. This approach is acceptable for both apple growers and crop protection compound manufacturers since they have both invested in their respective products. However, the excessive use of fungicides against apple scab has led to the emergence of resistant populations, which may compromise the management strategies in practice.

Protectant fungicides like Bordeaux mixture, dithianon, and captan are general enzyme inhibitors that affect primarily spore germination and have a very low risk of inducing resistance [3]. The main disadvantage of the protectant fungicides is that they rather wash off from the leaf surface quite easily with rain [4]; thus, they are usually applied either early in the season when there are only a few leaves present on the trees or when rain-free periods are expected [1,5]. Fungicides with systemic action are used: (a) mostly after rain events, when protectant fungicides are inadequate to protect the crop efficiently; (b) when the risk of primary infections is very high; and (c) to cover early plant growth in the spring [1,5]. Those fungicides are absorbed by the plant with varying levels of mobility and have a site-specific mode of action [6,7].

Compounds either inhibiting mitotic division (benzimidazoles) or involving cell membrane disruption (dodine) were exclusively applied since the 1970′s [8,9]. In the early 1980s, fungicides disrupting fungal sterol biosynthesis (demethylation inhibitors; DMIs) were applied widely to control many fungal diseases on tree fruits including apple scab; to date, myclobutanil and difenoconazole have been the most commonly used DMI fungicide against apple scab [10,11,12]. Over the last three decades, novel fungicides with different modes of action showing good efficacy against *V. inaequalis* were developed and introduced into practice. These fungicides were: cyprodinil and pyrimethanil (anilinopyrimidine class), inhibiting protein synthesis [13]; fludioxonil (phenylpyrrole class), interfering with the osmolarity glycerol pathway [14]; and pyraclostrobin and other quinone outside inhibitors (QoI class, strobilurins), affecting respiration by blocking the electron transport at the Qo site of the mitochondrial *bc*_1_ complex [7]. More recently, respiration inhibitors of the carboxamide class (2nd and 3rd generation SDHI’s), boscalid, fluopyram, and fluxapyroxad, were introduced, involving the succinate dehydrogenase (SDH) complex in the mitochondrial transport chain [15].

Fungicides that act primarily at single target sites, however, are characterized by an increased risk for the development of resistance [3]. Chemical control and resistance development in *V. inaequalis* towards different groups of fungicides were reviewed thoroughly by Cox [11]. In the United States of America, populations of the pathogen showing resistance to benomyl (benzimidazoles) and dodine were spread widely, resulting in the discontinued use of those fungicides [9,16,17]. Resistance of *V. inaequalis* to benzimidazoles was observed only seven years after their introduction [18], whereas it took more than 20 years before resistance to the DMI fungicides was well established in *V. inaequalis* populations [11]. Resistance to DMIs, including several cases of practical resistance in the field, has been reported worldwide [5,10,19,20,21]. By contrast, resistance to QoI fungicides in various plant pathogens, including *V. inaequalis,* was developed within 2 years after their introduction [17,22]. More recently, strains with reduced sensitivity to anilinopyrimidines (EC_50_ values for mycelial growth from 1 to 10 mg L^−1^ pyrimethanil and cyprodinil) were detected in *V. inaequalis* isolates in Italy [23] and New Zealand [24]. Strains of *V. inaequalis* resistant to SDHIs (boscalid) were found for the first time in apple orchards in Northern Italy [25].

The development of the resistance of *V. inaequalis* to chemical control agents needs continuous monitoring. According to the Food and Agriculture Organization (FAO), Greece produces approximately 300.000 metric tons of apples each year (ranking in 37th place worldwide). During the past few years, many Greek growers have raised some serious concerns about the efficacy of various formulations and mixtures of the DMI and QoI classes of fungicides and dodine after witnessing poor control against apple scab in practice. Hitherto, however, there is no available information on the fungicide sensitivity of *V. inaequalis* strains in Greece. Therefore, the present work was undertaken: (i) to demonstrate the sensitivity response of selected *V. inaequalis* isolates to eight fungicides of different chemical classes that are currently registered against apple scab, including fludioxonil (no data are available to the best of our knowledge); (ii) to determine the discriminatory doses for assessing the fungicide sensitivity by a simple method; and (iii) to investigate the existence and frequency of *V. inaequalis* strains with reduced sensitivity to all fungicides in three major apple-growing regions in Greece.

## 2. Materials and Methods

### 2.1. Fungicides

The following commercial fungicide formulations were supplied by their representative manufacturers and used in all tests: trifloxystrobin as Flint 50 WG (Bayer AG, Leverkusen, Germany); cyprodinil as Chorus 50 WG (Syngenta Crop Protection AG, Basel, Switzerland); dodine as Syllit 544 SC (Arysta LifeScience Benelux Sprl, Ougrée, Belgium); difenoconazole as Score 25 EC (Syngenta Crop Protection AG, Basel, Switzerland); boscalid as Cantus 50 WG (BASF SE, Ludwigshafen, Germany); fludioxonil as Geoxe 50 WG (Syngenta Crop Protection AG, Basel, Switzerland); dithianon as Delan 70 WG (BASF Agro BV, Arnhem, The Netherlands); and captan as Merpan 80 WG (Adama Makhteshim Ltd, Beer Sheva, Israel). The fungicides were dissolved in sterilized distilled water and stock solutions of 1000 mg L^−1^ active ingredient (a.i.) were prepared. To obtain the desired concentrations of each fungicide (mg L^−1^ of a.i. in growth medium), 10-fold serial dilutions were made from the stock solutions. To inhibit the alternative respiration function, the QoI fungicide trifloxystrobin was tested with the addition of 100 mg L^−1^ salicylhydroxamic acid (SHAM; Sigma-Aldrich, Darmstadt, Germany) to the medium. The final concentration of the solvent in the growth medium did not exceed 1% (*v*/*v*).

### 2.2. Isolates

During the 2020-2022 growing periods, *V. inaequalis* was isolated from the diseased leaves in major apple-growing areas in Greece located in three regions (Table 1). The identification of the pathogen was previously determined by Chatzidimopoulos et al. [26]. Diseased leaves bearing sporulating lesions were collected from twelve commercial apple orchards, located in the areas of Kastoria, Naousa, Alexandreia, and Zagora (Table 1). The orchards in Western and Central Macedonia (Kastoria, Naousa, and Alexandreia sites) were selected because of the poor apple scab control, which was reported over the last years. The fields in Zagora were selected at random since no fungicide resistance survey has been undertaken in this area. The orchards had been treated with DMIs at least three times each season (at least four in Kastoria), two times with dodine and two or three times with QoIs. However, the QoIs were also applied in May as mixtures with DMIs (most commonly as Flint max 75 WG; tebuconazole 50%, and trifloxystrobin 25%, Bayer AG, Leverkusen, Germany) to control powdery mildew, as well. The diseased leaves were transferred to the laboratory in individual paper bags to prevent cross-contamination.

*V. inaequalis* conidia were scraped from the sporulating leaf lesions and streaked onto a water agar medium in the form of a capital E [27]. After 24 h of incubation at 20 °C in the dark, the individual germinating conidia were transferred into glass tubes filled with a malt extract agar medium (MEA; Oxoid, Basingstoke, United Kingdom) and amended with 100 mg L^−1^ chloramphenicol. From each leaf, a single isolate was transferred. Colonies were incubated in a growth chamber at 20 °C in the dark for three weeks. For the fungitoxicity tests, fresh subcultures were made on MEA media and used as inocula. A total of four hundred and eighteen (418) single spore isolates were obtained.

### 2.3. Spore Germination and Mycelial Growth Assays

One hundred isolates originating from all the sampling areas were selected at random and subjected to a fungitoxicity analysis on conidial germ tube elongation and mycelial growth to calculate EC_50_ values. Volumes of 1 mL from stock fungicide solutions were added to 100 mL of autoclaved MEA media, cooled to approximately 45 °C and distributed in 6 cm Petri dishes. The final concentrations (10× dilution series) to the media were: 0.0001, 0.001, 0.01, 0.1, 1, 10, and 100 mg L^−1^. The dishes unamended with fungicides were used as controls. Additional control plates were amended with 100 mg L^−1^ of SHAM in order to check its inhibitory effect in tests with the QoI fungicide trifloxystrobin.

Conidia from 21-day-old cultures grown in glass tubes containing MEA media were used as inoculum. To obtain large scale conidia production, the fungus was grown on tubes containing MEA with each tube seeded with 0.3 mL of a conidial suspension [27]. By the use of a Neubauer improved hemocytometer (Preciss, Lancing, United Kingdom), aqueous spore suspensions containing 10^4^ spores per mL were prepared and three droplets (about 15 μL each), were pipetted on 6 cm Petri dishes with fungicide-amended media, replicated three times. The inoculum was spread evenly on the medium with a glass rod and the dishes were incubated in a growth chamber at 20 °C in the dark. After 36 h of incubation, the conidial germ tube elongation was recorded for at least 100 spores at each isolate/fungicide concentration. A conidium was considered germinated if the developing germ tube had the length of its longest diameter.

Mycelial discs of 4 mm in diameter were removed from the periphery of 21-day-old cultures grown on MEA media using a cork borer and aseptically transferred to the center of 6 cm Petri dishes with fungicide-amended media, prepared as on previous occasions. After 21 days of incubation at 20 °C in the dark, the radius of the colony (starting from the edge of the 4 mm plug) was measured with a digital caliber. Three replicates per each isolate/fungicide were made.

### 2.4. Screening for Resistance

On the basis of spore germination and mycelial growth tests, the discriminatory doses to assess the various sensitivity levels were determined and used for screening an additional three hundred and eighteen (318) isolates by the point inoculation method [28]. In brief, a drop (about 20 μL) of a dense spore suspension in sterile water (about 50,000 spores per mL) of each isolate was pipetted on pre-marked points on the periphery of Petri dishes containing MEA medium amended with the MICs for the sensitive isolates of each fungicide. After 10 days of incubation at 24 °C in the dark, the Petri dishes were assessed for fungal colonies visible to the naked eye.

The discriminatory doses for each fungicide used were: trifloxystrobin at 10 mg L^−1^ plus 100 mg L^−1^ SHAM; cyprodinil at 1 mg L^−1^; dodine at 1 mg L^−1^; difenoconazole at 0.1 and 1 mg L^−1^; boscalid at 1 mg L^−1^; fludioxonil at 1 mg L^−1^; and dithianon and captan at 10 mg L^−1^.

### 2.5. Data Analysis

The EC_50_ values for germ tube elongation and mycelial growth of each isolate/fungicide were calculated by the linear regression of the probit percentage of inhibition as a function of the log_10_ of different inhibitory fungicide concentrations. Statistical analysis was performed with the ARM software (GDM solutions Inc., Brookings, SD, USA). The Super-Plots-Of-Data application [29] was used to plot the EC_50_ values into categories and compare the means distribution. The Shapiro-Wilk test was used to detect the normality of the data and Welch’s t-tests were used to compare differences (p-values) among the data sets (EC_50_ values) of the sensitivity groups.

## 3. Results

In the germ tube elongation assays conducted in fungicide-amended media, conidia had germination rates of over 80%, which were similar to the germination rates observed in the control plates, unless stated otherwise. Depending on the isolate, the length of the germ tube on the control plates ranged from 170 to 280 μm at 36 h after inoculation with an average value of 200–220 μm. The respective colony growth (radius) after 21 days of incubation ranged between 7 and 12 mm (Figure 1). Compared to the control plates, the addition of SHAM caused a 15% reduction on either germ tube length or mycelial growth on all isolates tested.

### 3.1. Characterization of Fungicide Sensitivity Profiles

#### 3.1.1. Sensitivity to Trifloxystrobin

Two distinguishable sensitivity groups of EC_50_ values were formed. The EC_50_ values of six isolates for germ tube elongation and mycelial growth were in the narrow range of 0.1755 to 0615 mg L^−1^ and 0.1245 to 0.339 mg L^−1^, respectively (Figure 2a). These isolates failed to form colony at the concentration of 10 mg L^−1^ and were characterized as sensitive to the QoIs.

In contrast, the EC_50_ values for the germ tube length of ninety-four isolates were in a range from 15.21 to over 100 mg L^−1^. The respective EC_50_ values for colony formation ranged from 5.467 to over 100 mg L^−1^ (Figure 2b). Those isolates were separated from the most sensitive isolate by a resistance factor (RF; highest EC_50_ value/lowest EC_50_ value) of over 569.8 for germ tube elongation and over 803.2 for mycelial growth, respectively, and characterized as resistant to the QoIs. The fungicide concentration of 10 mg L^−1^ plus 100 mg L^−1^ SHAM, which completely inhibited the growth of sensitive isolates, was adopted as a discriminatory dose for detecting resistant strains to trifloxystrobin.

#### 3.1.2. Sensitivity to Cyprodinil

The EC_50_ values for germ tube length and colony growth ranged from 0.1615 to 7.8145 mg L^−1^ and from 0.1752 to 9.8974 mg L^−1^, respectively. The most resistant strain was separated from the most sensitive by a RF value of 48.39 for germ tube length and 56.49 for mycelial growth, respectively. Among the isolates tested, twenty-three had EC_50_ values for both germ tube elongation and mycelial growth ranging from 0.1615 to 0.8055 mg L^−1^ cyprodinil (Figure 3). These isolates failed to form a colony at the concentration of 1 mg L^−1^ of cyprodinil and were characterized as sensitive to anilinopyrimidines. The rest of the isolates had EC_50_ values for mycelial growth over 1 mg L^−1^ cyprodinil and were characterized as moderately resistant to anilinopyrimidines.

The mycelial growth of the sensitive isolates had lower EC_50_ values on average compared to the germ tube elongation. The isolates with EC_50_ values for germ tube length between 0.8 and 1 mg L^−1^ had EC_50_ values for colony formation close to 1 mg L^−1^ and gave a sparse colony formation in point inoculation assays at the discriminatory dose of 1 mg L^−1^ cyprodinil. All isolates germinated at the concentration of 1 mg L^−1^, while eighty-two isolates failed to germinate at the concentration of 10 mg L^−1^. However, eighteen isolates with EC_50_ values over 50 mg L^−1^ for germ tube elongation had 20% of their conidia germinated at the concentration of 10 mg L^−1^ with a germ tube length of 80 μm and developed a colony with a 4–6 mm radius. The concentration of 1 mg L^−1^ cyprodinil was adopted as the inhibitory dose for differentiating two sensitivity levels to this anilinopyrimidine fungicide.

#### 3.1.3. Sensitivity to Dodine

The effect of this fungicide against both stages of development of the fungus was similar. The EC_50_ values of the isolates scattered within a range from 0.0141 to 9.9918 mg L^−1^ dodine and two distinguishable sensitivity groups were formed (Figure 4). Sixty-four isolates had EC_50_ values for both the germ tube length and mycelial growth ranging from 0.0141 to 0.8254 mg L^−1^ having a peak (34 isolates) at the concentrations close to 0.30 mg L^−1^ of dodine (Figure 4). The majority of these isolates failed to germinate at the concentration of 1 mg L^−1^ dodine while a limited mycelial growth was observed (radius of 2–3 mm). However, in 28 of those isolates, 5% of the conidia were able to germinate at this concentration, having a germ tube length below 50 μm.

Thirty-six isolates had EC_50_ values for a germ tube length from 0.9155 to 8.8974 and for mycelial growth from 1.0161 to 9.9918 mg L^−1^ (Figure 4). Among these isolates, 32 (with EC_50_ values below 5 mg L^−1^) were not able to have their conidia germinated at the concentration of 10 mg L^−1^, but were able to form a colony with a limited radius of 2–3 mm. On four isolates (the EC_50_ values for both germ tube elongation and mycelial growth were between 5 and 10 mg L^−1^), 30% of the conidia were germinated at a concentration of 10 mg L^−1^, having 8 μm length, and 8% of the conidia were even able to germinate at a concentration of 100 mg L^−1^, having a length of 50 μm. A limited mycelial growth with 2–3 mm radius was also observed at a concentration of 100 mg L^−1^. The most resistant strain was separated from the most sensitive one by an RF value of 419.69 for germ tube growth, while for mycelial growth the most resistant strain was separated from the most sensitive isolate by an RF value of 708.64. The concentration of 1 mg L^−1^ dodine was defined as the inhibitory dose for differentiating two sensitivity levels.

#### 3.1.4. Sensitivity to Difenoconazole

The EC_50_ values for mycelial growth were distributed in a relatively wider range compared to germ tube elongation assays (Figure 5). The EC_50_ values for germ tube length and mycelial growth ranged from 0.1009 to 11.6541 and from 0.0183 to 11.2745 mg L^−1^ difenoconazole, respectively. Figure 5 shows that two groups of EC_50_ values were formed for germ tube elongation and three for mycelial growth; the latter consists of the primary growth stage affected by this class of fungicides. The most resistant strain was separated from the most sensitive by an RF value of 115.5 for germ tube length and 616.09 for mycelial growth, respectively.

The isolates of the most sensitive group had EC_50_ values for germ tube elongation that ranged from 0.1009 to 0.3655 mg L^−1^ difenoconazole. Those isolates had EC_50_ values for colony formation in the range of 0.0183 to 0.0496 and failed to form a colony at a concentration of 10 mg L^−1^. Isolates with EC_50_ values for mycelial growth in the range of 0.1 to 0.7 mg L^−1^ were able to form a colony with a 2–4 mm radius at 10 mg L^−1^ whereas isolates with EC_50_ values above 10 mg L^−1^ were able to develop a colony with a 2 mm radius at 100 mg L^−1^ difenoconazole. The respective EC_50_ values for germ tube growth of the resistant groups ranged from 1.1755 to 11.6541 mg L^−1^ difenoconazole. None of the spores germinated at a concentration of 100 g L^−1^. On the basis of both fungitoxicity assays, a dose of 0.1 mg L^−1^ difenoconazole was selected as the primary inhibitory dose for differentiating two sensitivity levels (moderate resistance). Additionally, a dose of 1 mg L^−1^ was used to check higher levels of resistance on point inoculation assays.

#### 3.1.5. Sensitivity to Boscalid

Two distinguishable sensitivity groups of EC_50_ values for both spore germination and mycelial growth were formed (Figure 6). Boscalid was found more effective against germ tube length. The EC_50_ values of ninety-five isolates for germ tube elongation and mycelial growth ranged from 0.0161 to 1.0727 and from 0.1531 to 7.802 mg L^−1^ boscalid, respectively. These isolates failed to germinate (20% germinated, but the length of the germ tube was lower than the spore size) at a concentration of 1 mg L^−1^. Furthermore, none of these isolates formed a colony at a concentration of 10 mg L^−1^ boscalid.

The other five isolates had EC_50_ values for germ tube elongation from 2.9417 to 7.8745 mg L^−1^ and for mycelial growth from 12.9417 to 37.2145 mg L^−1^ boscalid (Figure 6). These isolates had 50–70% of their conidia germinated and formed a colony with a 5 mm radius at a concentration of 10 mg L^−1^; thus, they classified as moderately resistant to boscalid. A concentration of 1 mg L^−1^ was adopted as the discriminatory dose for detecting resistant strains to boscalid. The resistance factor for germ tube elongation ranged from 2.74 (lowest EC_50_ of the resistant/highest EC_50_ of the sensitive strains) to 489.01 (highest EC_50_ of the resistant/ lowest EC_50_ of the sensitive strains) and for mycelial growth from 1.66 to 243.07.

#### 3.1.6. Sensitivity to Fludioxonil

This fungicide was more effective on the germination stage of the fungus with EC_50_ values in the range of 0.0015 to 1.0061 mg L^−1^. The sensitivity on mycelial growth was lower with EC_50_ values ranging between 0.1531 and 21.147 mg L^−1^ (Figure 7). At a concentration of 0.1 mg L^−1^ fludioxonil, eighty-three of the isolates either did not germinate (10%) or 5–50% of their conidia germinated with germ tube length below 100 μm. These isolates failed to form a visible colony at a concentration of 1 mg L^−1^ in the point inoculation assays and were characterized as sensitive to fludioxonil.

Seventeen isolates had at least 50% of their conidia germinated at a concentration of 0.1 mg L^−1^ with a germ tube length of over 100 μm. At four of these isolates, the conidia grew uninhibited at this concentration and they were even able to germinate (with rates of 15–50%) at a concentration of 1 mg L^−1^ fludioxonil with germ tube lengths varying from 70 to 120 μm. The mycelial growth of those isolates ranged from 8.7875 to 21.147 mg L^−1^ (Figure 7b). The most resistant strain was separated from the most sensitive one by RF values of 670.73 and 138.125 for germ tube length and mycelial growth, respectively. On the basis of both fungitoxicity assays, a concentration of 1 mg L^−1^, which completely inhibited the growth of the sensitive strains, was assigned as the discriminatory threshold dose to assess the sensitivity response to fludioxonil.

#### 3.1.7. Sensitivity to Dithianon and Captan

The EC_50_ values for germ tube elongation varied within the ranges of 0.2727 to 9.9847 and 0.913 to 14.8745 mg L^−1^ for dithianon and captan, respectively (Figure 8). The EC_50_ values for mycelial growth were over 100 mg L^−1^ for both fungicides. The conidia of most isolates were not able to germinate at a concentration of 10 mg L^−1^ of dithianon or captan. Although the most resistant strain was separated from the most sensitive by an RF of 36.61 for dithianon and 16.29 for captan on the spore germination assays, we were rather reluctant to characterize them as moderately resistant due to the multi-site activity of those fungicides. However, on the point inoculation assays, many isolates were able to germinate (at least 50% germination with germ tube length varying from 50 to 100 μm) and form a visible colony at a concentration of 10 mg L^−1^ of dithianon and captan, and we recorded this incidence as a slight reduction in sensitivity.

### 3.2. Survey on Fungicide Resistance

Fungicide resistant isolates were detected in sampling cites with commercial apple orchards. The majority of those isolates had a high resistance to the QoIs followed by a moderate resistance to the anilinopyrimidines (Figure 9). Resistance to difenoconazole and dodine was also detected on all sampling sites, but at various frequencies.

However, the resistant isolates to difenoconazole were detected only in the orchards in Western (8 out of 52%) and Central Macedonia (2 out of 34%). The isolates moderately resistant to boscalid and fludioxonil were detected at lower frequencies in the orchards of Western and Central Macedonia, but none were detected in Zagora (Figure 8). Finally, a reduced sensitivity to dithianon and captan was observed in a small percentage of isolates originating from all sampling sites.

Among the four hundred and eighteen (418) isolates, only three isolates (0.7%) were sensitive to all fungicides. Two isolates (0.5%) were resistant to all fungicides including dithianon and captan. A comparison of the EC_50_ values of a sensitive and a multiple resistant isolate is given in Table 2. Most isolates were found to be resistant or moderately resistant to three, four, or five fungicides with frequencies of 38.8, 27.5, and 16.2%, respectively. Finally, isolates either resistant or moderately resistant to one (2.2%), six (5.2%), and seven fungicides (2.2%) were detected at lower frequencies.

## 4. Discussion

The emergence of resistant variants in fungal populations subjected to selective pressure by exposure to fungicides is the result of an evolutionary process [3]. To differentiate the resistant (or less sensitive) variants of *V. inaequalis* over a baseline population towards different groups of fungicides, many researchers have defined critical concentrations in which the growth of the most sensitive variants is inhibited. The values of those concentrations may vary based on the populations’ sensitivity; however, this is not necessarily related to the practical resistance in the field [10]. In the present assay, it has been demonstrated that *V. inaequalis* strains have shown various levels of resistance to all fungicides including the multi-site inhibitors, dithianon and captan. Since there were no previous studies on baseline sensitivities and fungicide resistance surveys in Greece, the differentiation was made on the established populations. The adjustment of the point inoculation method on the behavior of the pathogen in the discriminatory doses that were defined by two stages of development (germ tube elongation and mycelial growth), was found to be a simplified procedure to scan a large number of isolates for resistance in many different fungicides.

The QoI fungicides, trifloxystrobin and kresoxim-methyl, were first introduced into practice around two decades ago [11]. From the results in the present study, it becomes apparent that the extensive use of those fungicides has led to an appearance of practical resistance in the field. Trifloxystrobin (Flint 50 WG, Bayer AG, Leverkusen, Germany) is now registered against apple scab and powdery mildew on apples at doses of up to 10 g per 100 liters of water, which corresponds to 50 mg of active ingredient (a.i.) per liter (mg L^−1^). In all major apple growing areas in Greece, the resistance to QoIs is now well-established in *V. inaequalis* populations with frequencies in the range of 89 to 92%. Most likely, this kind of resistance is qualitative and controlled by the G143A mutation on the *cytb* gene, as reported in previous studies [20,30,31]. However, strains with EC_50_ values ranging from 0.1245 to 0.615 mg L^−1^, which were characterized as sensitive in the present assay, are considered, in other studies, as resistant based on EC_50_ values below 0.01 mg L^−1^ of the baseline population [22]. Those strains may have quantitative resistance which occurs below the point at which the alternative respiration pathway becomes activated and move towards immunity [11]. Despite the fact that the farmers were advised to replace the QoIs with fungicides from other chemical groups in spray programs against scab, as in Uruguay [10], the use of those fungicides, either combined with fungicides from other chemical groups or in mixtures, will continue as they were found very effective against powdery mildew. Therefore, the status is not expected to alter in the next few years since the qualitative resistance of the QoIs was found to be stable for prolonged periods [31].

Larsen et al. [24], in a survey in New Zealand’s apple orchards during the 2011–2012 growing period, reported the occurrence of moderately resistant *V. inaequalis* strains to anilinopyrimidines with EC_50_ values greater than 0.5 mg L^−1^. Likewise, Fiaccadori [23] observed poor control of apple scab in orchards that had been treated with pyrimethanil with EC_50_ values for resistant strains of up to 18.3 mg L^−1^. In the present assay, strains of *V. inaequalis* moderately resistant (MIC of 1 mg L^−1^) to the anilinopyrimidines were detected in high frequencies on all sampling sites. However, in an experimental trial site, we have demonstrated that those strains were efficiently controlled at a rate of 50 g of Chorus 50 WG (cyprodinil 50%; Syngenta Crop Protection AG, Basel, Switzerland) per 100 L of water, which corresponds to 250 mg a.i. L^−1^ [1]. Interestingly, Kunz et al. [32] found no shift in anilinopyrimidine sensitivity after 43 treatments in tests in vivo. The anilinopyrimidine class of fungicides was first registered for use into practice in Greece back in 2003. However, unlike the QoIs, all these years most of the farmers limited the use of anilinopyrimidines to only a few per season according to the instructions on the label and thus, their good efficacy against apple scab has been retained. An increased frequency of moderately resistant strains to anilinopyrimidines in Central Macedonia might be explained by the fact that those strains were subjected to additional selection pressure from sprays with anilinopyrimidine fungicides against *Monilinia* spp. in neighboring orchards with peaches.

Previous formulations of dodine were associated with an increased risk of fruit russeting when the water in the tank mix was cold [10]. Hence, the inclusion of dodine in spray programs against apple scab was limited to only a few per season. In the present assay, the moderately resistant group of strains was separated from the sensitive one by an average resistance factor of 9.68 in both germ tube elongation and mycelial growth assays. However, the range of the resistance factor in mycelial growth (highest EC_50_/lowest EC_50_) reached up to 651.15. The mean EC_50_ values of over 1 mg L^−1^ for the resistant population of *V. inaequalis* are in line with previous reports from Canada [33], New Zealand [34], and the USA [6]. However, we observed that a few strains were able to form a colony even at a concentration of 100 mg L^−1^. Moreover, Yoder and Klos [35] in a study back in 1976, reported the growth of *V. inaequalis* in a solution containing 300 mg L^−1^ of dodine. Despite the relatively high frequency (16–33%) of moderately resistant strains in all sampling sites, it is unlikely that a resistance of practical importance will arise. Dodine (as Syllit 544 SC, Arysta LifeScience Benelux Sprl, Ougrée, Belgium) is currently registered at 83–625 mL 100 L^−1^ of water (451.52-3400 mg a.i. L^−1^), which is far above the MIC of 1 mg L^−1^. In a field trial, Chatzidimopoulos et al. [1] did not observe any failure in control of apple scab when dodine was applied at a dose of 1250 mL per ha.

The resistance response to DMI fungicides is typically considered incremental, quantitative, or dose-dependent, presumably controlled by more than one gene and may occur after several years of intensive use [10,11]. This is further supported by the fact that later studies reported increased EC_50_ values for difenoconazole after many years of use [20,21] compared to earlier studies performed several years ago [10,12,36]. In orchards located in Kastoria, difenoconazole was applied at least four times each season for over a decade at a maximum rate of 20 mL 100 L^−1^ (50 mg a.i. L^−1^) Score 25 EC (Syngenta Crop Protection AG, Basel, Switzerland) because the farmers relied heavily on the efficacy of this fungicide against both apple scab and powdery mildew; as a result, eight percent of the total isolates were found resistant to difenoconazole. However, the concurrent presence of sensitive and moderately resistant isolates on the same site provides further evidence for the quantitative type of resistance. Schnabel and Jones [37] have found that an overexpression of the *CYP51A1* gene, which encodes the target protein 14α-demethylase, is an important mechanism of resistance in some field resistant strains of *V. inaequalis*, but other mechanisms of resistance may also exist. However, previous studies have shown that no cross-resistance has been observed between the DMI fungicides myclobutanil and difenoconazole [11,12]. Moreover, Cox [11] determined that difenoconazole has a higher intrinsic activity against *V. inaequalis* than myclobutanil when both compared at 0.1 mg L^−1^ a.i. In a field trial for the control of apple scab in Kastoria in 2022, we concluded that the new member of the DMIs, mefentrifluconazole (Revyona SC, BASF Agro BV, Arnhem, The Netherlands), is a worthy replacement of difenoconazole in spray programs (unpublished data).

To date, there is little information on the resistance to boscalid and fludioxonil. Approximately 5% of the isolates that originated from orchards in Central Macedonia were moderately resistant to boscalid and 13% to fludioxonil. The respective frequencies in Kastoria were lower most likely because those isolates were subjected to additional pressure by neighboring sprays in peaches against *Monilinia* spp., similar to the anilinopyrimidines. Interestingly, there were no *V. inaequalis* strains resistant to these two classes of fungicides in Thessaly. In a previous study, the populations between Macedonia and Thessaly were found to be genetically distant [26]. The mixtures of boscalid and pyraclostrobin (Signum 26,7/6,7 WG), and fludioxonil and cyprodinil (Switch 25/37.5 WG, Syngenta Crop Protection AG, Basel, Switzerland), are among the most frequently used fungicides against monilia in stone fruits. In Kastoria, boscalid is mainly applied in a mixture with pyraclostrobin as Bellis 25,2/12,8 WG (BASF SE, Ludwigshafen, Germany) in summer to prevent late infections because the orchards’ microclimates are affected by the lake Orestiada and pinpoint scab symptoms may appear in fruits due to a high relative humidity. The EC_50_ values from 12.94 to 37.21 mg L^−1^ for the mycelial growth of the moderately resistant strains to boscalid are in line with those reported in a study for a resistance monitoring program in Italy [25]. In *B. cinerea*, resistance to boscalid is reported to be related to several point mutations within the *sdh* gene-encoding proteins of complex II [38]; however, further research is needed to characterize the genetic background of the resistance to SDHIs in *V. inaequalis*. To the best of our knowledge, resistance to fludioxonil (moderate) in *V. inaequalis* is reported for the first time worldwide. Fludioxonil is now registered against several pathogens in apples either solo as Geoxe 50 WG (Syngenta Crop Protection AG, Basel, Switzerland) or in a mixture with cyprodinil as Switch 25/37.5 WG (Syngenta Crop Protection AG, Basel, Switzerland). Kilani and Fillinger [39] reported that a high resistance to fludioxonil has not existed among plant pathogenic fungi worldwide for over 30 years now, most probably due to the fitness penalty, and this status is not expected to change in the future. Hence, the data suggests that this decrease in fludioxonil sensitivity in *V. inaequalis* is not expected to compromise the use of phenylpyrrole fungicides. However, in *B. cinerea*, strains with a moderate resistance to fludioxonil have been associated with an underlying mechanism of multidrug resistance (MDR), which involves mutations leading to overexpression of ABC (ATP binding cassette) and MFS (major facilitator superfamily) transporters [40]. Further research is under progress to clarify if a similar mechanism exists in *V. inaequalis* strains that would explain this reduced sensitivity in fungicides from unrelated chemical groups.

Dithianon and captan are multi-site inhibitors with certain activity against thiol groups in proteins and peptides [41,42]. This group of fungicides is considered by the Fungicide Resistance Action Committee (FRAC) as low risk for developing resistance. Both fungicides affect the spore germination in *V. inaequalis* and are used in spray programs in a preventive way. Dithianon is considered a vital tool against apple scab and can be used up to ten times in a season. In the present assay, we used the MIC of 10 mg L^−1^ to differentiate the less susceptible strains, but we avoided characterizing them as moderately resistant due to the multi-site activity of those fungicides. However, on point inoculation assays, 4 to 11% of the isolates in all areas were able to form colonies similar to the control. The EC_50_ values for dithianon and captan were found elevated compared to previous studies in Italy and Serbia [36,42]. Even though the recommended rates are high above the MIC of 10 mg L^−1^, we certainly observed a slight reduction in sensitivity, which may play a role in the poor control of apple scab on some orchards. Fiaccadori [42] also found a reduction in dithianon activity in tests in vivo and speculated that this loss of sensitivity might be caused by temporary modifications in the target of this fungicide due to increased inoculum pressures.

## 5. Conclusions

In conclusion, *V. inaequalis* isolates with various levels of resistance to commonly used site-specific and muti-site inhibitors were selected in major apple growing areas in Greece, under the pressure of the excessive use of fungicides for many years. This selection was more prominent in an area near a lake in which the production of mass *V. inaequalis* inoculum is favored by the microclimate every year. Isolates exhibiting high resistance to trifloxystrobin and difenoconazole, and moderate resistance to cyprodinil, dodine, boscalid, and fludioxonil, were found for the first time in Greece. A small percentage of the isolates were also found less sensitive to captan and dithianon. It is more likely that a combination of all the above (mass inoculum, presence of resistant population, and high humidity) with additional difficulties to intervene in time because of the rain events are factors that contribute to poor control of apple scab. However, it must be pointed out that this poor control of apple scab is limited to only a few locations. It has been previously shown that all the moderately resistant isolates can be controlled effectively by using the fungicides on the highest recommended rates [1]. Since the QoIs will continue to be a part of the spraying programs against apple scab, farmers should either reduce the use of difenoconazole or replace it with another member of the DMIs, such as mefentrifluconazole, to efficiently protect their final product.

## Figures and Tables

**Figure 1 microorganisms-10-02354-f001:**
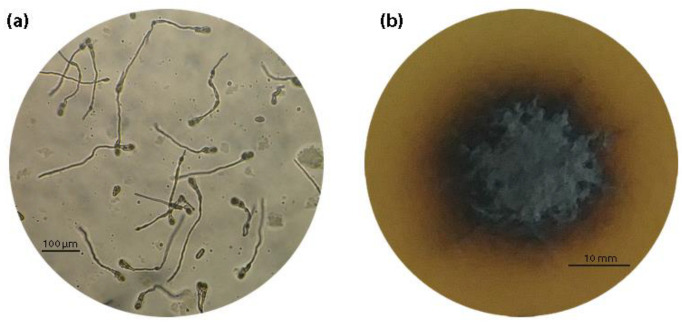
The development of different *V. inaequalis* growth stages on control plates with MEA: (**a**) germ tube elongation at 36 hours after inoculation with aqueous spore suspension containing 10^4^ spores per mL; and (**b**) mycelial growth at 21 days after inoculation with a 4 mm mycelial disc. The images were taken with Nikon Eclipse Ei upright microscope (Nikon Europe BV, Amstelveen, The Netherlands) using 10× (**a**) and 4× (**b**) plan achromat objective lens.

**Figure 2 microorganisms-10-02354-f002:**
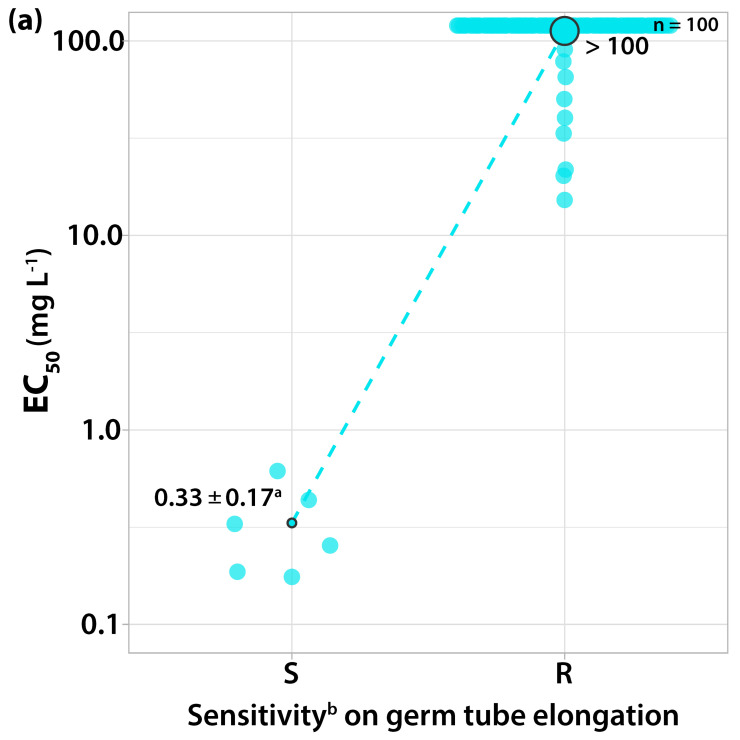
Frequency of sensitivity distribution for one hundred *V. inaequalis* isolates to trifloxystrobin on (**a**) germ tube elongation and (**b**) mycelial growth. The mean EC_50_ values for the resistant population of *V. inaequalis* were significantly higher (*p* < 0.05) than the mean EC_50_ values of the sensitive population on both tests according to the Welch’s t-test. ^a^ Average values of sensitive isolates ± standard deviation; circle size reflects the population number. ^b^ Sensitivity groups: S, sensitive; R, resistant.

**Figure 3 microorganisms-10-02354-f003:**
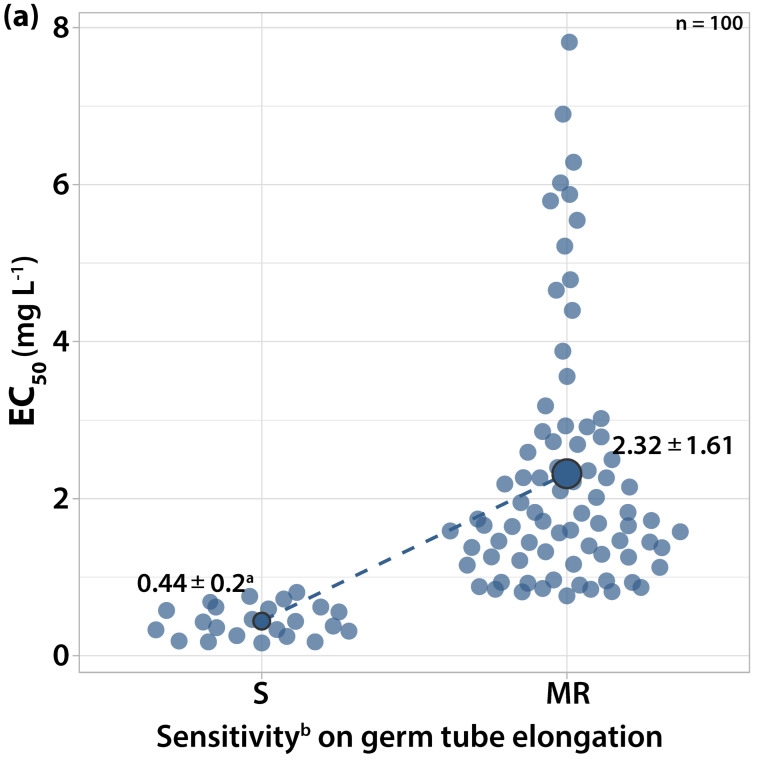
Frequency of sensitivity distribution for one hundred *V. inaequalis* isolates to cyprodinil on (**a**) germ tube elongation and (**b**) mycelial growth. The mean EC_50_ values for the moderately resistant population of *V. inaequalis* were significantly higher (*p* < 0.05) than the mean EC_50_ values of the sensitive population on both tests according to the Welch’s *t*-test. ^a^ Average values of sensitivity groups ± standard deviation; circle size reflects the population number. ^b^ Sensitivity groups: S, sensitive; MR, moderately resistant.

**Figure 4 microorganisms-10-02354-f004:**
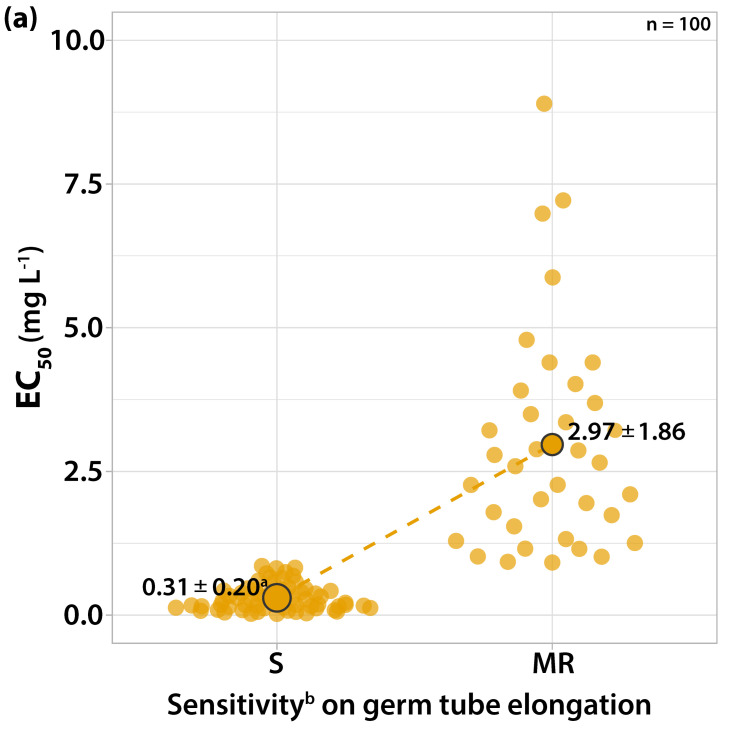
Frequency of sensitivity distribution for one hundred *V. inaequalis* isolates to dodine on (**a**) germ tube elongation and (**b**) mycelial growth. The mean EC_50_ values for the moderately resistant population of *V. inaequalis* were significantly higher (*p* < 0.05) than the mean EC_50_ values of the sensitive population on both tests according to the Welch’s *t*-test. ^a^ Average values of sensitivity groups ± standard deviation; circle size reflects the population number. ^b^ Sensitivity groups: S, sensitive; MR, moderately resistant.

**Figure 5 microorganisms-10-02354-f005:**
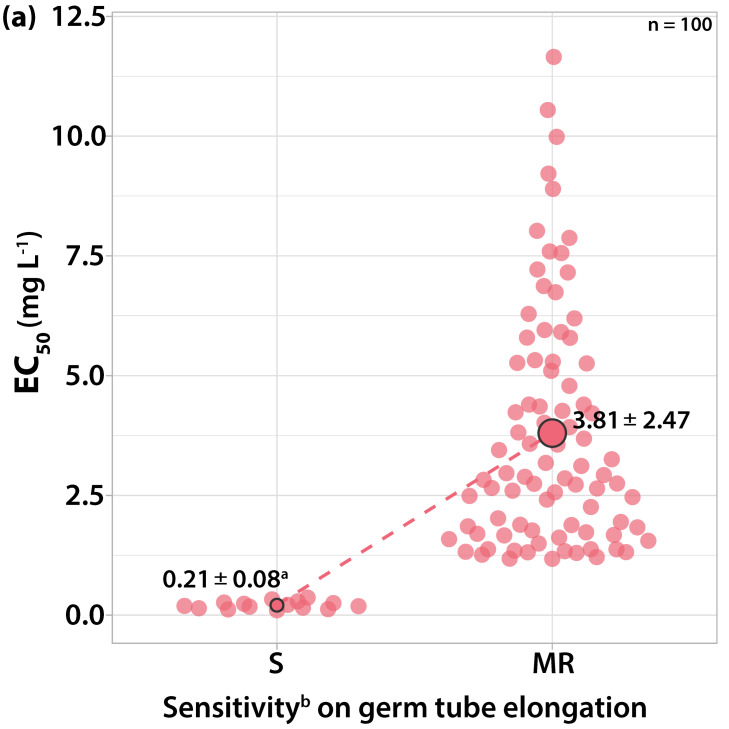
Frequency of sensitivity distribution for one hundred *V. inaequalis* isolates to difenoconazole on (**a**) germ tube elongation and (**b**) mycelial growth. The mycelial growth stage was found to be more sensitive than the germ tube elongation during the bioassays with difenoconazole. Since the DMI fungicides primarily affect the mycelial growth of the fungi, the concentrations of 0.1 and 1 mg L^−1^ were used to discriminate the sensitivity levels on point inoculation assays. The mean EC_50_ values for the moderately resistant or resistant population of *V. inaequalis* were significantly higher (*p* < 0.05) than the mean EC_50_ values of the sensitive population on both tests according to the Welch’s *t*-test. ^a^ Average values of sensitivity groups ± standard deviation; circle size reflects the population number. ^b^ Sensitivity groups: S, sensitive; MR, moderately resistant; R, resistant.

**Figure 6 microorganisms-10-02354-f006:**
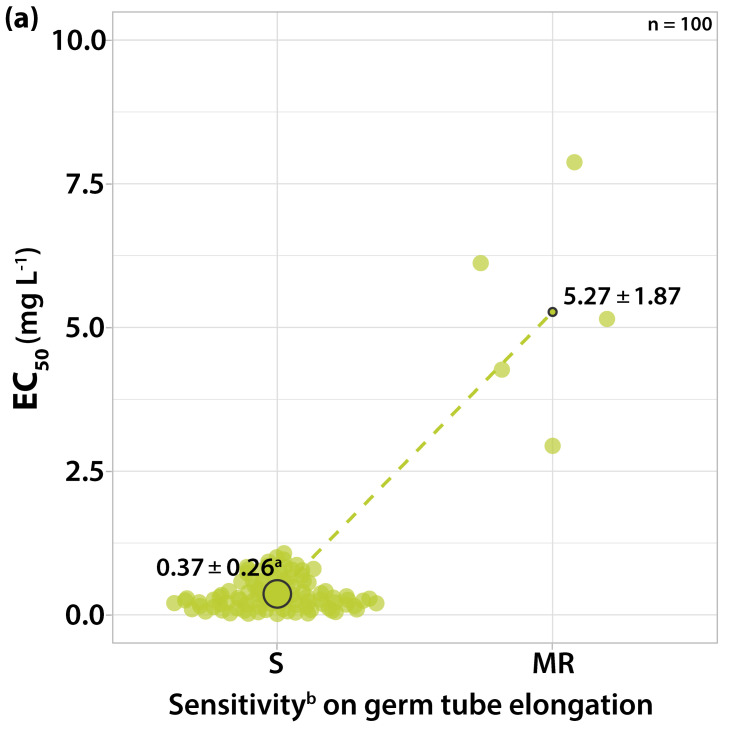
Frequency of sensitivity distribution for one hundred *V. inaequalis* isolates to boscalid on (**a**) germ tube elongation and (**b**) mycelial growth. The mean EC_50_ values for the moderately resistant population of *V. inaequalis* were significantly higher (*p* < 0.05) than the mean EC_50_ values of the sensitive population on both tests according to the Welch’s *t*-test. ^a^ Average values of sensitivity groups ± standard deviation; circle size reflects the population number. ^b^ Sensitivity groups: S, sensitive; MR, moderately resistant.

**Figure 7 microorganisms-10-02354-f007:**
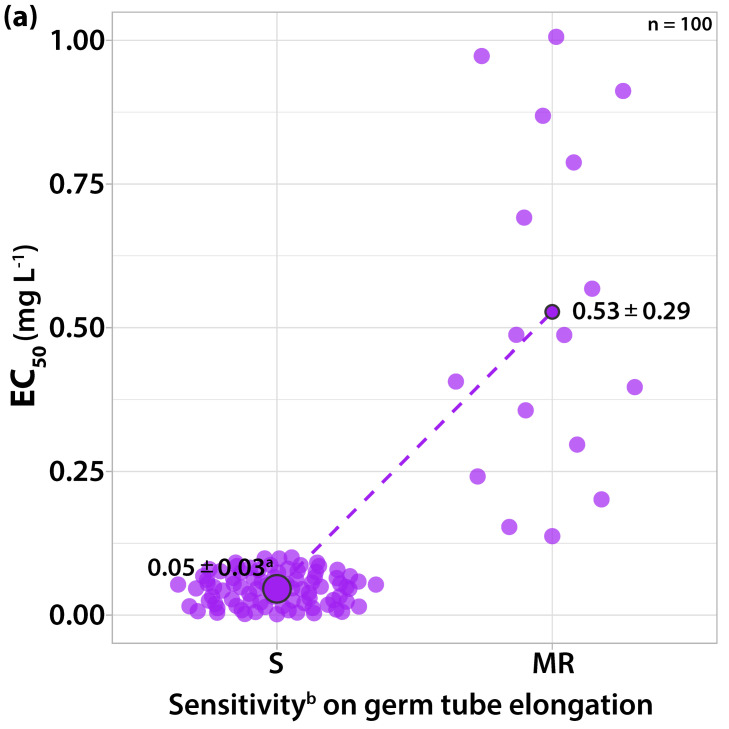
Frequency of sensitivity distribution for one hundred *V. inaequalis* isolates to fludioxonil on (**a**) germ tube elongation and (**b**) mycelial growth. The mean EC_50_ values for the moderately resistant population of *V. inaequalis* were significantly higher (*p* < 0.05) than the mean EC_50_ values of the sensitive population on both tests according to the Welch’s *t*-test. ^a^ Average values of sensitivity groups ± standard deviation; circle size reflects the population number. ^b^ Sensitivity groups: S, sensitive; MR, moderately resistant.

**Figure 8 microorganisms-10-02354-f008:**
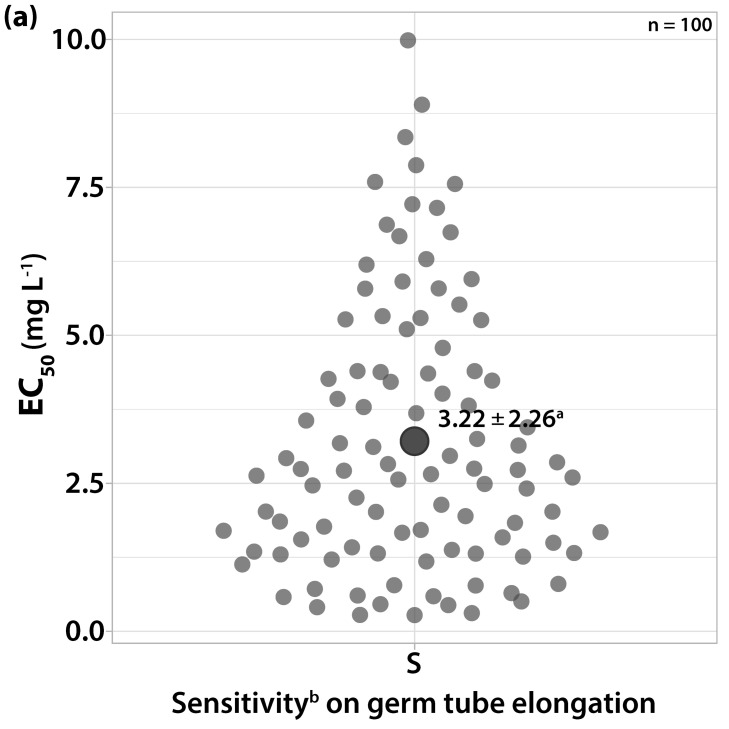
Frequency of sensitivity distribution for one hundred *V. inaequalis* isolates on germ tube elongation to (**a**) dithianon and (**b**) captan. ^a^ Average values of sensitivity groups ± standard deviation; circle size reflects the population number. ^b^ Sensitivity group: S, sensitive.

**Figure 9 microorganisms-10-02354-f009:**
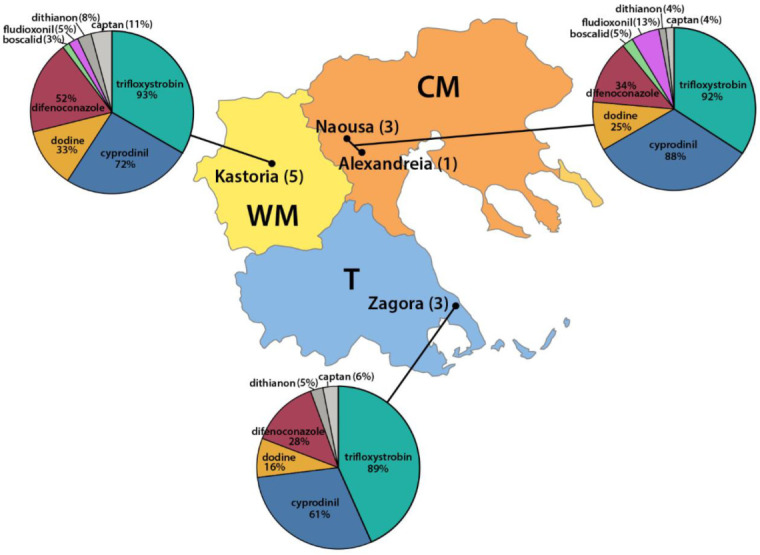
Frequency distribution of resistant *V. inaequalis* isolates to eight fungicides of unrelated chemical groups. Sampling sites in the regions of Western Macedonia (WM), Central Macedonia (CM), and Thessaly (T).

**Table 1 microorganisms-10-02354-t001:** Origin of *V. inaequalis* isolates used.

Orchard Number	Site (Region ^a^)	Apple Cultivar	Number and Code of Isolates
1	Kastoria (WM)	Gala SchniCo Red	48 (KG)
2	Kastoria (WM)	Super Chief	57 (KT 1-57)
3	Kastoria (WM)	Super Chief	36 (KT 58-93)
4	Kastoria (WM)	Fuji Kiku	29 (KF 1-29)
5	Kastoria (WM)	Fuji Kiku	51 (KF 30-80)
6	Naousa (CM)	Scarlet spur	35 (NT)
7	Naousa (CM)	Super Chief	30 (GT)
8	Naousa (CM)	Fuji Kiku	27 (RT)
9	Alexandreia (CM)	Pink Lady	41 (VT)
10	Zagora (T)	Scarlet spur	19 (ZG 1-19)
11	Zagora (T)	Scarlet spur	25 (ZG 20-44)
12	Zagora (T)	Scarlet spur	20 (ZG 45-64)

^a^ Regions of Western Macedonia (WM), Central Macedonia (CM), and Thessaly (T).

**Table 2 microorganisms-10-02354-t002:** Comparison of the germ tube elongation and mycelial growth of a sensitive and a multiple resistant *V. inaequalis* isolate.

	EC_50_ for Germ Tube Elongation			EC_50_ for Mycelial Growth		
Fungicides	Wild-Type Isolate KT10	Multiple Resistant Isolate VT18	RF	Wild-Type Isolate KT10	Multiple Resistant Isolate VT18	RF
Trifloxystrobin ^1^	0.1495	>100	>668.90	0.3219	>100	>310.66
Cyprodinil	0.1727	6.8340	39.57	0.2919	9.6899	33.20
Dodine	0.048	5.9083	123.09	0.0558	4.3968	78.80
Difenoconazole	0.1248	5.7317	45.93	0.0183	1.8744	102.43
Boscalid	0.0849	2.1692	25.55	0.8954	30.62	34.20
Fludioxonil	0.0033	0.2532	76.72	0.4586	25.96	56.61
Dithianon	0.4572	8.8974	19.46	>100	>100	-
Captan	1.2987	10.7727	8.29	>100	>100	-

^1^ SHAM was added at 100mg L^−1^; RF: EC_50_ value of the resistant isolate divided by EC_50_ value of the wild-type isolate.

## Data Availability

No applicable.

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
