# Peer review of "Detection of Venturia inaequalis Isolates with Multiple Resistance in Greece"

_microorganisms, 2022, doi:10.3390/microorganisms10122354_

Round 1

Reviewer 1 Report

The manuscript is well written scientifically and systematically; the article investigates the extent of Venturia inaequalis resistance emergence as a result of the usage of several fungicides in Greece. The article will aid local apple growers in developing management plans for apple scab.

The manuscript can be accepted after incorporating the following information:

1.    Describe the methodology utilized to identify V. inaequalis isolates.

2.      It would be preferable to add a picture of the pathogen and the infected leaf.

3.      A different heading should be used for the conclusion.

Reviewer 2 Report

General comments:

The manuscript entitled "Detection of Venturia inaequalis isolates with multiple resistance in Greece" attempts to measure the resistance of Venturia inaequalis isolates to eight chemical fungicides used in apple cultivation.

Although the article is well written and relevant to farmers in the study area, I feel that the article is not novel or original enough to be published in Microorganisms. This article is useful to implement training programs and the proper use of fungicides in the farmers of these three communities; but the international relevance of this article is limited and of local interest.

In general, I am not totally convinced by the applied methods. The study-dependent variables (germ tube elongation and mycelial growth of each isolate/fungus) are adequate to measure sensitivity to toxicity towards different fungicides, but it is not recommended to infer resistance towards them. In this sense, a genetic mapping study was not carried out to search for genes involved in resistance to fungicides, such as the sdhB gene. It is recommended to adjust the results obtained for sensitivity to toxicity. In addition, you can consult the following work: Molecular Characterization of the sdhB Gene and Baseline Sensitivity to Penthiopyrad, Fluopyram, and Benzovindiflupyr in Venturia inaequalis. https://doi.org/10.1094/PDIS-12-15-1512-RE

In the methodology section, the manuscript needs to corroborate the causal agent of the disease, nothing is mentioned about an identification and classification of native isolates, this can be by classical or molecular taxonomic route, and at least one isolate must be deposited in the Genbank database and place the accession number in the manuscript, this to confirm that it really is Venturia inaequalis. In this sense, the authors are recommended to confirm the pathogenicity of at least one characteristic isolate through Koch's postulates.

Principal component analysis or multivariate analysis is recommended to denote the area with the least sensitivity to fungicides. This would further highlight the results obtained. In addition, it could correlate sensitive, moderately toxic, and highly toxic isolates.

Comments by section:

Introduction

Add data to the writing:

The importance of apple production in Greece and the place it occupies in production worldwide.

Of the three sites evaluated, mention if they are the most important producers and their contribution to national production.

Mention the variety of apple cultivation.

Mention and explain why there are resistance to the toxicity of the different types of fungicides.

Line 94. Change resistance for toxicity.

Methodology

Line 22, attach a citation confirming the information about the 120-year driving period.

No classical or molecular taxonomic identification was performed. It is suggested to corroborate the species through specific molecular markers (ITS 1 and 4) and 1-alpha (tef1) of at least one characteristic isolate and enter its Genbank accession number.

Pathogenicity testing is recommended. See following article: Koopman, T. A., Meitz-Hopkins, J. C., Tobutt, K. R., Bester, C., & Lennox, C. L. (2022). Pathogenicity and virulence of south African isolates of Venturia inaequalis. European Journal of Plant Pathology, 164(1), 45-58.

Results

It is recommended to put photographs for the elongation of the germ tube and the most representative mycelial growth in the eight fungicides used.

Line 365. Change subtitle for "Distribution of resistance to toxicity to different fungicides"

Line 384-390. Apparently, they are conclusions of the work, it is recommended to eliminate.

Table 1 is not mentioned in the text. In addition, it is suggested to carry out a statistical analysis to confirm differences.

It is recommended to carry out a principal component analysis or a multivariate analysis to denote the area with the highest toxicity to fungicides. This would further highlight the results obtained. In addition, it could correlate sensitive, moderately toxic, and toxic isolates. Tests with genetic markers are lacking to confirm if there is expression of genes related to resistance. It is recommended to handle the term toxicity, high toxicity, or low toxicity, which is more focused on the variables established in the investigation.

Discussion

The regulation of fungicides in the country of origin is not mentioned, it is recommended to mention this regulation and it would be interesting to know if applications in the crop have increased in recent years.

It is recommended to comment on the type of agronomic management and against the disease for each zone and indicate which of the three zones has the lowest incidence of disease or the lowest accumulated toxicity with respect to production.

Conclusion

Line 518, put subtitle for the conclusion. Review the forms of shipment of the magazine.

Line 532-534. The authors state that they found isolates with high resistance to trifloxystrobin and difenoconazole and moderate resistance to cyprodinil, dodine, boscalid and fludioxonil without having performed any genomic mapping. However, the scope of the investigation with two variables studied does not confirm this assertion. It is recommended to eliminate this type of comments and focus on the results obtained, which is toxicity.

Line 534-535. This statement is correct according to the study, sensitivity was seen according to the toxicity of the fungicides. In this sense, the conclusions must be rewritten.

Line 541. In conclusions, no type of reference to the work is put, eliminate reference.

Bibliography

Adequate.

Reviewer 3 Report

Chatzidimopoulos et al., report the results of fungicides (trifloxystrobin, difenoconazole, cyprodinil, dodine, difenoconazole, boscalid, fludioxonil, dithianon, captan) screenings of Venturia inaequalis (the causal agent of apple scab) isolates collected from orchards throughout Greece.  The authors present evidence of varying degrees of resistance to multiple fungicides from in vitro colony growth assays and germ tube growth elongation. This is apparently the first published evidence of fungicide resistance in V. inaequalis in Greece, which is an alert to producers and plant pathologists to expect diminishing effectiveness of apple scab control through fungicides.

These results are meaningful for apple producers in Greece and the potentially provide baseline data against which the strength of fungicide resistance can be compared over time (as a means of monitoring and possible modification of management approaches).

I have some concerns about the current version of the manuscript that I hope the authors can address.

I could not understand the methods of the assays themselves and their subsequent statistical analyses.  This is because both aspects were not fully explained in a manner that could be replicated. 

First, it is unclear whether the authors performed a true dilution series (which is typical for quantifying the level of fungicide resistance, often indexed by an EC50 value – the concentration of fungicide at which colony growth/reproductive rate is reduced by half compared to the same isolate in the absence of the fungicide.  The authors’ description of the methods suggest that a dilution series was used but they invoke a MIC (Minimum Inhibitory Concentration) value somehow.  Typically, each isolate is characterized and a value is presented for each isolate to characterize the population of isolates.  But, from Figures 1-7, it appears that the isolates were not characterized separately but rather as two groups of isolates that were categorized based on a method (perhaps a post-hoc assignment from their EC50 calculations?).  How the individual isolates were evaluated experimentally should be explained in detail and should be complemented and linked specifically to the statistical analysis itself.  Both aspects should be explained with enough detail and rationale that the experiment could be reasonably replicated.  I think that the authors can attain the necessary detail to make experiment repeatable by other plant pathologists, and it would greatly improve the manuscript and give the reader confidence in the authors’ results and interpretation.  Presently, these methodological aspects are unclear and I had to make numerous assumptions about the methods which may or may not be correct.   

Second, the isolates appear to be anonymous.  This is an important issue because the reader does not know whether the isolates are comprised of a mixture of a few clones, an equal mixture of clones and unique genotypes, or mostly unique genotypes. Typically, in fungicide resistance studies, the isolates are genotyped, either as a means to screen for mutations that are known to confer fungicide resistance and whether the isolates are clones. If the authors are able to genotype their isolates, this would be consistent with the fungicide resistance literature and enable the authors to more thoroughly evaluate the potential genetic mechanisms underlying the observed resistant phenotype.  Presently, the study is highly restricted by the lack of genotype information.

Round 2

Reviewer 2 Report

The manuscript entitled "Detection of isolates of Venturia inaequalis with multiple resistance in Greece" has improved significantly and the authors explain the doubts they had about the methodological part, stating that it is a continuing work and attaching relevant information in order to have a better comprehension.

It is difficult to determine when an organism presents resistance to a chemical agent or when it initiates a tolerance mechanism, which is why further research should be carried out in this line of research and present more evidence for future research.

Line 94. Change resistance for toxicity. Answer: The term toxicity is usually applied in insects. The alternative would be reduced sensitivity by we use resistance (in various levels) to highlight the difference with the wildtype isolates.

The toxicity does not necessarily apply to insects, check the following research as an example: https://doi.org/10.3390/jof7080598

Line 22, attach a citation confirming the information about the 120-year driving period. Answer: I am sorry, I cannot understand.

I mean that the author should put a reference to the period of the place selected for his poor control of apple scab.

Reviewer 3 Report

This is a much improved manuscript and I appreciate the steps that the authors took in their revisions. 

There are still a couple of concerns, but I am sure that the authors can address these concerns.

Minor concern

Lines 210-212 (this is a new sentence in the revision).  I do not understand the last half of the sentence from an analysis standpoint (the authors are using a t-test to assess p-values?).  Please reword this sentence to clearly explain the analysis logic.  It is possible that this will be more than one sentence.

Major concern. 

Thank you to authors for providing the genotyping results in the table, this is helpful.  However, the isolate analysis is still based on a mixture of genetically anonymous clones as the manuscript is presented.  Because the authors have genotyped these isolates and report them in another manuscript, they should be able to link the genotype and phenotype directly in this present manuscript for Figures 1-7.  The way to do this, which I feel is far more faithful and rigorous with respect to the initial data is to represent each of the 12 genotypes listed in Table (of the revised manuscript) with a separate symbol/color for each genotype.  This would enable the reader to better visualize the genotype data overlaid on the phenotype data and perhaps some important patterns will emerge relative to within genotype variation and positioning within the range of the fungicide resistance continuum.  I think this would be very useful and potentially be of interest to a broader readership.
